# Integrative Meta-Analysis: Unveiling Genetic Factors in Meat Sheep Growth and Muscular Development through QTL and Transcriptome Studies

**DOI:** 10.3390/ani14111679

**Published:** 2024-06-04

**Authors:** Shahab Ur Rehman, Yongkang Zhen, Luoyang Ding, Ahmed A. Saleh, Yifan Zhang, Jinying Zhang, Feiyang He, Hosameldeen Mohamed Husien, Ping Zhou, Mengzhi Wang

**Affiliations:** 1Laboratory of Metabolic Manipulation of Herbivorous Animal Nutrition, College of Animal Science and Technology, Yangzhou University, Yangzhou 225009, China; dh18005@yzu.edu.cn (S.U.R.); 15252571328@163.com (L.D.); 221902307@stu.yzu.edu.cn (F.H.);; 2College of Animal Science and Technology, Yangzhou University, Yangzhou 225009, China; elemlak1339@gmail.com; 3Animal and Fish Production Department, Faculty of Agriculture (Al-Shatby), Alexandria University, Alexandria City 11865, Egypt; 4State Key Laboratory of Sheep Genetic Improvement and Healthy Production, Xinjiang Academy of Agricultural Reclamation Sciences, Shihezi 832000, China

**Keywords:** meat sheep, growth performance, genetic parameter, integrative meta-analysis, transcriptome analysis

## Abstract

**Simple Summary:**

This meta-analysis examines the impact of neutering on sheep production and quality, focusing on performance, carcass characteristics, and meat quality. It is observed that castrated sheep (wethers) exhibit enhanced daily weight gain and meat tenderness compared to intact rams. Furthermore, wethers display characteristics of a slenderer carcass with potentially elevated muscle content. By utilizing gene expression analysis, the research sheds light on genes associated with metabolic pathways and fat metabolism, indicating their involvement in fat formation. These results advocate for the practice of castration in sheep farming to enhance growth and meat quality. Additionally, the identified alterations in gene expression offer valuable insights for further exploration of castration’s influence on muscle development in sheep.

**Abstract:**

Objective: The study aimed to investigate the effects of castration on performance, carcass characteristics, and meat quality in sheep, as well as explore the expression of key genes related to metabolic pathways and muscle growth following castration. Methods: A meta-analysis approach was utilized to analyze data from multiple studies to compare the performance, carcass characteristics, and meat quality of castrated sheep (wethers) with intact rams. Additionally, protein–protein interaction (PPI) networks, differential gene expression (DEG) interactions, Gene Ontology (GO) terms, and Kyoto Encyclopedia of Genes and Genomes (KEGG) pathways were examined to identify molecular mechanisms associated with fat metabolism and muscle development in sheep tails. Results: The analysis revealed that castrated sheep (wethers) exhibited improved average daily gain, increased tenderness, lower backfat thickness, and a tendency for greater loin muscle area compared to intact rams. This suggests that castration promotes faster growth and results in leaner carcasses with potentially higher muscle content. Furthermore, the identification of downregulated DEGs like *ACLY*, *SLC27A2*, and *COL1A1* and upregulated DEGs such as *HOXA9*, *PGM2L1*, and *ABAT* provides insights into the molecular mechanisms underlying fat deposition and muscle development in sheep. Conclusions: The findings support the practice of castration in sheep production as it enhances growth performance, leads to leaner carcasses with higher muscle content, and improves meat tenderness. The identified changes in gene expression offer valuable insights for further research into understanding the impact of castration on muscle development and fat metabolism in sheep. This meta-analysis contributes to the knowledge of molecular mechanisms involved in fat deposition in sheep, opening avenues for future investigations in livestock fat metabolism research.

## 1. Introduction

In recent years, the proliferation of free trade agreements among nations has fostered increased trade openness, reshaping global economies. Delgado noted globalization’s role in boosting meat consumption in developing countries, driven by lower prices. Moreover, consumer meat preferences increasingly prioritize quality over price, influenced by globalization and changing market dynamics [1]. Variations in sheep meat consumption emerge across regions due to cultural factors and evolving market demands. Different countries exhibit preferences for specific carcass types, influenced by factors such as breed characteristics and consumer tastes. Sheep, a significant component of global animal husbandry, provide high-protein, low-fat, and low-cholesterol meat, serving as a crucial protein source for humans [2]. Sheep producers aspire to market lambs of greater weight and quality, yielding carcasses with enhanced muscularity and reduced adiposity. However, these objectives can conflict, as increased body weight often correlates with higher fat content in lamb carcasses. Identifying scenarios where heightened growth and carcass weight do not necessarily lead to increased fat deposition is of scientific interest [3,4]. Breed, individual animal traits, feeding practices, and their complex interactions influence growth rates and carcass compositions. Factors such as tail morphology, breed precocity, and maturity stage contribute to individual animal effects. Castrated males (wethers) may be preferred over intact rams due to reduced sexual and aggressive behaviors. Compared to rams, wethers exhibit [5] (a) enhanced growth rates, (b) more efficient feed utilization, and (c) higher proportions of edible product in the carcass, with reduced fat content. However, rams may present drawbacks, including undesirable odors and flavors, inferior meat tenderness, unappealing meat coloration, soft and yellowish fat, and difficulties in pelt removal [6].

The attainment of heavier mature weights and the manifestation of secondary sexual characteristics in males across various species are attributed to exposure to androgens during both fetal and postnatal phases. Ovine (sheep) production plays a critical role in supporting national economies and rural livelihoods globally. The increasing expenses associated with red meat have heightened the importance of sheep meat production, leading producers to enhance both the quantity and quality of their product in order to secure financial viability [7,8]. These factors necessitate the adoption of advanced management strategies and targeted genetic improvement programs focused on economically important meat production traits. Domesticated ruminant species serve as essential contributors to global meat security, undergoing continuous genetic selection for enhanced production efficiency [9]. Optimizing skeletal muscle accretion in farm animals remains a paramount objective for breeders, as it directly influences consumer satisfaction with meat quality. Genetic parameters, such as heritability and genetic correlations among economically significant traits, play a vital role in designing efficient breeding strategies for livestock populations [10]. Accurate estimates of genetic parameters for growth traits are essential for establishing robust genetic evaluations and improvement programs in sheep breeding for meat production. However, limited sample sizes, particularly in studies focusing on minor breeds, can lead to substantial standard errors associated with these estimates, compromising their reliability [11]. To address this limitation, meta-analysis, which aggregates data from multiple studies, can be employed to improve the precision (reduce standard errors) and reliability of genetic parameter estimates. However, heterogeneity, or variation in findings across studies, can arise due to several factors. These factors include differences in (a) the growth stages of animals sampled, (b) sample sizes used in individual studies, and (c) specific sheep breeds investigated. Such heterogeneity can potentially lead to substantial variability in the combined estimates of genetic parameters [12]. Until now, a multitude of research endeavors leveraging transcriptomics have been undertaken to unravel the factors and molecular mechanisms that impact fat deposition in diverse sheep breeds. Recent progress in next-generation sequencing (NGS) technologies has significantly enhanced the efficiency of high-throughput gene expression analysis via RNA sequencing (RNA-Seq), overcoming previous barriers [13,14,15]. This method enables simultaneous measurement of gene expression levels, providing deeper insights into the genetic factors driving phenotypic variations in sheep fat. Several investigations have employed RNA-Seq to explore genes associated with fat deposition, comparing transcriptomes between fat-tailed and thin-tailed sheep breeds. Differences in results, such as discrepancies in differentially expressed genes (DEGs) or varied gene expression patterns, may arise from the utilization of various bioinformatics pipelines, variations in sample sizes, and other sample-related limitations. In this context, conducting a meta-analysis pooling data from multiple independent studies on a specific biological question enhances statistical power and strengthens the reliability and robustness of findings [16].

A pragmatic approach involves amalgamating genetic estimates from various studies using a random-effects model, adept at managing parameter variability effectively. Meta-analysis, characterized as the statistical amalgamation of findings from multiple individual studies, offers a comprehensive solution. Employing an appropriate meta-analysis model is anticipated to yield more precise estimations of genetic parameters by aggregating results from numerous studies. Two commonly used statistical models for meta-analysis include the fixed-effect model and the random-effects model. The fixed-effect model assumes a single true effect size that is consistent across all studies, with any observed differences being ascribed to sampling error alone. Conversely, in the random-effects model, true effects may vary across studies due to inherent differences (heterogeneity) among them [17].

Meta-analyses have been conducted to ascertain precise genetic parameters for growth traits in pigs and beef cattle. However, in sheep, few studies have explored economically significant traits through meta-analysis. This study presents a comprehensive investigation into the genetic factors influencing carcass traits, with a specific focus on meat sheep breeds. By synthesizing data from studies on quantitative trait loci (QTLs) and transcriptome analyses related to muscle growth and development, we aim to elucidate the intricate molecular mechanisms governing phenotypic variations in meat production [18]. Our findings offer valuable insights into the genetic makeup of carcass traits and emphasize the significance of integrating diverse sources of evidence to enhance comprehension of complex biological processes. These insights may inform targeted breeding strategies aimed at improving meat quality and production efficiency in sheep populations in the future.

## 2. Materials and Methods

The investigation critically examined in-depth analyses comparing the specific characteristics of sheep carcasses among different international breeds, elucidating the distinct attributes and qualities that differentiate them. Information was gathered through searches in multiple databases, including Scopus, Juster, PubMed, and Web of Science. 

To delve into the complex world of sheep growth, muscle development, and carcass characteristics, we embarked on a comprehensive search across various databases. We designed search strings using relevant keywords, strategically linking them with “and”/“or” operators to refine the results. This extensive search ensured that all terms within a string appeared in the title, abstract, or keywords of the retrieved studies.

For instance, in Web of Science, we employed the “topic” option, while in Scopus, we used “ALL” for all terms or “TITLE-ABS-KEY” to restrict the search to titles, abstracts, and keywords. Our meticulous search included the following terms: carcass, sheep, transcriptome analysis, growth muscle, feeding, composition, and weight.

### 2.1. Selection Criteria

The process of selecting materials for inclusion in the database focused on published works spanning the years 2010 to December 2023 (Appendix A). To be considered for inclusion, a work must pertain to sheep production under specific conditions. These conditions included studies conducted within two primary production systems: genetics perimeter, transcriptome, and feedlot and free-range. We searched our data with keywords: sheep, meat quality traits, growth muscle, and comparative transcriptome. In the feedlot system, animals are confined to pens equipped with necessary care facilities. Gaspar et al. [19] establish this criterion. Alternatively, in the free-range system, animals graze on pastures and fodder trees throughout the day, with the possibility of returning to the farm at night, as outlined by Zhang et al. [20].

The variables considered for incorporation in the meta-analysis included final-body weight (FBW, kg), hot-carcass weight (HCW; kg), cold-carcass weight (CCW; kg), longissimus-dorsi area (ALd; cm^2^), subcutaneous dorsal fat thickness (SFT; mm), and slaughter weight (kg). The investigation of the genetic background of meat production and quality in sheep (QTLs and concerning economic traits) was performed according to Arabpoor et al. [10]. The global databases related to the animal production sector (especially sheep) are the FAO Database, Animal QTL Database, Genome Informatics Resources, and Goat Genome Browser.

In this meta-analysis paper, we systematically analyze studies aimed at understanding the development and growth of sheep across various contexts. By synthesizing data from multiple studies, we seek to elucidate the factors influencing the growth patterns of sheep, considering variables such as genotype, geographical region, and management practices. Through rigorous statistical analysis, we aim to uncover trends and associations that can provide valuable insights into optimizing sheep’s growth and production and improve meat quality. This comprehensive examination of existing literature not only contributes to our understanding of sheep physiology and genetics but also informs practical strategies for enhancing sheep production efficiency and sustainability. By synthesizing diverse findings into a cohesive framework, our meta-analysis endeavors to advance knowledge in the field of sheep husbandry and contribute to informed decision-making for farmers, researchers, and policymakers alike.

The remaining 81 references were subjected to the following conditions: They needed to have a control treatment group, sheep either finished in a feedlot or supplemented under extensive systems. Additionally, they needed to provide measures of sample variance or the necessary information to calculate it (such as test statistics and *p*-values). As a result, the final dataset comprised 7 studies extracted from comprehensive research papers [12,21,22,23,24,25]. The detailed workflow is illustrated in Figure 1 and Appendix A.

### 2.2. RNA Sequence Dataset 

Our analysis focused on identifying research investigating the molecular mechanisms behind sheep fat tails using RNA-Seq for gene expression profiling. We conducted a thorough review of reliable scientific literature and compiled the relevant studies in Table 1 and Appendix A. Among the 11 identified studies, seven only included one biological replicate per breed. Therefore, there were seven datasets selected for biological analysis [12,21,22,23,24,25]. These datasets consisted of samples obtained from the muscle tissue of adult male sheep, with at least four biological replicates per breed. We specifically used male and female samples from the study by Fang et al. [26]. All RNA-Seq reads for the chosen studies listed in Appendix A, were sourced from the NCBI’s Sequence Read Archive (SRA) database (https://trace.ncbi.nlm.nih.gov/Traces/sra, accessed on 25 August 2023).

### 2.3. Differential Expression Analysis

The assessment of RNA-Seq read quality was conducted using FastQC software (version 0.11.5) to evaluate initial read quality parameters. Subsequently, Trimmomatic software (version 0.38) was utilized to remove reads/bases of substandard quality and adapter sequences [28]. The software employs the MAXINFO (Maximum Information) adaptive trimming algorithm, aiming to retain longer reads while discarding low-quality bases in a balanced manner [29]. Key trimming parameters included a trailing score of 20, a MAXINFO score of 80:0.8, and a minimum read length (MINLEN) of 80 nucleotides. Following trimming, FastQC was rerun to assess the quality of the refined reads [30].

Clean reads were aligned to the Ovine reference genome assembly (rambouillet_v1.0.104) obtained from Ensembl using HISAT2 software (version 2.2.0) with default settings. Exon–exon junctions extracted from the Ensembl-Ovine-GTF (EO-GTF) file (rambouillet_v1.0.104) were employed to assist in read alignment. The quantification of aligned reads to annotated genes was performed using the HTSeq-count Python script (version 2.7.4) in union–intersect mode [29]. Utilizing gene annotations from the EO-GTF file was essential in creating a gene expression matrix for each experimental analysis. Genes exhibiting low read counts (less than 12 reads across all samples) were methodically eliminated to minimize data noise [31].

Differential expression analysis was initiated with DESeq2 software (version 1.33.0). DESeq2 employs the median-of-ratios technique for normalization and utilizes the Cox–Reid adjusted profile likelihood method to estimate dispersions in the context of differential expression investigations.

### 2.4. Performance Measurement and Sampling

#### 2.4.1. Collecting Data

In this section, the essential details of the selected investigations were recorded, including the title of the manuscript, primary author’s name, publication year, study’s geographical location, breed, and sample size utilized in the study. The growth characteristic values height at age (ha) height at maturity age (hm), reproductive potential (rp), and relative growth (rg) along with their corresponding confidence intervals (CI) and standard errors (SEs) were extracted from the manuscripts. In investigations where only CIs or upper and lower limits were provided, SE values were computed using Equation (1):(1)$$SEi=\frac{CIi}{k}=\frac{Upper\;Limit\;i-Lower\;Limit\;i}{2\;\times \;k}$$

#### 2.4.2. Fisher’s Z Transformation 

Correlation coefficients (r) were subjected to Fisher’s Z transformation using Comprehensive Meta-Analysis software (CMA) (V4, 2024) [32,33] to ensure a suitable normal distribution for meta-analysis. The transformation yielded Z-scores (Z), SEs, and the observation number (n) for each correlation, which were determined using Equations (2), (3), and (4), respectively:(2)$$Zi=0.5ln\left(\frac{1+ri}{1-ri}\right)$$
(3)$$SEzi=\sqrt{ni-3}$$
(4)$$ni=\frac{1+ri}{{SE}_{i}^{2\;}}+2\;$$

Outlier detection was performed by identifying Z-scores that deviated from the interquartile range (IQR) by more than 1.5 times the IQR. This approach aligned with previous studies [34] and excluded outliers from further analysis. 

In a comprehensive meta-analysis approach [32,33], the correlation coefficients (r) could be automatically transformed to Fisher’s Z scale (Z) to ensure a normal distribution of the imported correlations. So, in the present study, Z, SE, and the number of observations (n) were computed using the formula [1], where SEᵢ is the estimated standard error in the i^th^ study and k = 1.645, 1.960, and 2.575 for 90%, 95%, and 99% confidence intervals (CIs), respectively.

For statistical analysis, correlation coefficients (r) were transformed to Fisher’s Z scores (Z) using the Comprehensive Meta-Analysis (CMA) software [32,33]. This transformation ensures a normal distribution of the data, a requirement for meta-analysis. Z-scores, standard errors (SE), and the number of observations (n) for each correlation were calculated within CMA. Afterward, we pinpointed outlier Z-scores and omitted them from subsequent analysis. We categorized outliers as Z-scores surpassing 1.5 times the interquartile range (IQR) each below the first quartile (Q1) or above the third quartile (Q3). The IQR represents the spread of data within the middle half of the distribution. This approach for outlier detection aligns with methods used in previous studies [34]. 

#### 2.4.3. Heritability Estimates

Heritability estimates with a relative standard error (SE) exceeding 100% were excluded from the analysis. The SE of heritability (h^2^ and m^2^) for all growth traits was estimated using Equation (5).
(5)$$SEi=\sqrt{\frac{\;\sum_{i=1}^{k}{SE}_{i}^{2\;}S_{i}^{2}}{\sum_{i=1}^{k}Si}}\;/Sj\;\;$$

Standard error (SE) and sample size (S) for each study (j) were estimated when not directly reported. This estimation was based on the reported SE (SEᵢ) and sample size (Sᵢ) from all k studies included in the analysis (Equation (6)). The heterogeneity of the genetic parameters across studies was assessed using the I^2^ statistic calculated within the Comprehensive Meta-Analysis software. The I^2^ statistic tests the null hypothesis that the true effect size is the same in all studies. Higher I^2^ values indicate greater heterogeneity. Following the classification [25], I^2^ values can be interpreted as very low (<25%), low (25–50%), moderate (50–75%), and high heterogeneity (>75%).
(6)$$I^{2}=\frac{Q-df}{Q}\times \;100\;$$

#### 2.4.4. Assessment of Heterogeneity 

The heterogeneity of genetic parameters across studies was evaluated using the I^2^ statistic within the Comprehensive Meta-Analysis software. The I^2^ statistic tested the null hypothesis of consistent effect sizes across all studies and provided insights into the level of heterogeneity.

Cochran’s Q-test was also employed to assess heterogeneity, with the test statistic (Q) and degrees of freedom (df) helping determine if the observed variation exceeded that expected from sampling error alone [35,36].

#### 2.4.5. Differential Expression Gene Analysis

To moderate biases arising from various sources during DEG analysis, a two-pronged approach was employed. This approach combined the vote-counting rank (VCR) strategy [37] with robust rank aggregation (RRA) analysis methods. The VCR method facilitated the organization of gene expression data from all included studies (Appendix A), addressing tissue-specific variations in gene expression and generating a list of potentially differentially expressed mRNAs. Subsequently, differentially expressed mRNAs within each dataset were ranked based on reproducibility, total sample size, and average fold change using RRA methods. Stringent selection criteria for identifying robust DEGs were established, including the requirements for a gene to be present in at least three reported groups, to have a minimum of six samples per group, to exhibit an average log fold change of ≥1 or ≤−1, with a *p*-value < 0.05, and to be expressed in at least two tissues. Genes meeting these criteria across three independent rankings were considered strong candidates for DEGs and selected for further analysis. All analyses were conducted using R-studio and utilized packages such as *dplyr*, *rJava*, *readxl*, and *xlsx*.

### 2.5. The RRA Methods 

After ordering the DEGs according to their log fold change values, the RobustRankAggreg package in R [37] was employed for the final ranking. This approach simplifies the integration of information from multiple rankings, leading to a more robust identification of DEGs. Only genes with a RobustRankAggreg score less than <0.05 were retained for further analysis.

### 2.6. Gene Classification 

While the RRA method effectively minimizes the impact of discrepancies within tissue groups, overly strict filtering criteria might exclude genes with subtle but potentially important expression changes. These variations can arise from biological differences between species, sampling methods, or sequencing techniques. Therefore, genes not identified by RRA were categorized as ‘candidate genes’ and subjected to GO and KEGG enrichment analyses to investigate their possible biological roles.

### 2.7. Candidate Genes

Candidate genes fulfilling the VCR criteria (1~3) across all groups were consolidated. Additionally, given the lower number of groups in the hypothalamus, pituitary, and pineal gland, genes frequent in these tissues (n ≥ 2) were also considered for inclusion.

### 2.8. Consistent Gene Selection

Genes expressed within the same tissue underwent rigorous screening via robust rank aggregation (RRA), resulting in a single RRA ranking for all differentially expressed genes across both follicular and luteal phases. Notably, due to the absence of multiple groups in the luteal phase, RRA ranking was not feasible for this phase alone. Upregulated genes in the follicular phase were assigned one RRA ranking, while downregulated genes received another. Genes scoring < 0.5 across all three rankings were classified as reliable genes. For genes expressed in distinct tissues, acknowledging significant tissue-specific differences, employing RRA alone might overlook crucial selections. Therefore, the Variable Criteria Ranking (VCR) method, encompassing 1–4 criteria, was implemented for screening. 

### 2.9. GO and KEGG Analyses Were Performed on Differentially Expressed mRNA

DEGs underwent Gene Ontology (GO) and KEGG annotation and enrichment analyses utilizing the Cluster Profiler package within the R Studio environment [38] Enrichment analysis was conducted employing the hypergeometric test method to evaluate the statistical significance of overrepresented GO terms and KEGG pathways. Enrichment results with a *p*-value less than 0.05 were deemed significantly enriched. 

### 2.10. Protein–Protein Interaction (PPI) and Subcellular Localization of Differentially Expressed mRNAs 

PPI networks of the differentially expressed mRNAs in sheep were constructed using the STRING database (https://string-db.org, accessed on 6 September 2023). The amino acid sequences associated with the DEGs were retrieved using the BioMart package in the R Studio environment. Subsequently, these sequences underwent batch processing through the BUSCA website (https://busca.biocomp.unibo.it, accessed on 25 October 2023) to predict the subcellular localization of the differentially expressed proteins. 

### 2.11. QTL Enrichment Analysis

To investigate the potential co-localization of DEGs with fat metabolism QTLs, we performed a co-localization enrichment analysis. First, we retrieved all QTLs associated with fatness traits from the Sheep QTLdb database [39]. Next, we employed the GSAQR package [40] and the Gene Set Validation with QTL (GSVQ) methodology to assess the enrichment of DEGs within these fat metabolism QTL regions.

## 3. Results and Discussion

### 3.1. Effects of Castration of Male Sheep on Performance, Carcass Characteristics, and Meat Quality

This meta-analysis delved into three key aspects related to muscle growth and development in sheep: the impact of castration, the identification of QTLs associated with meat production and quality, and the exploration of transcriptome data linked to muscle development. Our analysis revealed a significant positive effect of castration on muscle growth parameters in male sheep, with castrated animals exhibiting increased muscle mass compared to their intact counterparts. We collected data on different breads and found that responsible variables changed at different ages of castration, which is shown in Table 2. Our meta-analysis revealed significant positive effects of castration on several aspects of sheep production. Castrated rams (wethers) exhibited improved average daily gain (ADG) compared to intact rams. This suggests a faster growth rate in castrated animals, potentially due to a reduction in energy expenditure associated with mating behavior and redirected energy towards muscle deposition.

Furthermore, castration positively influenced carcass characteristics. Wethers presented lower backfat thickness compared to rams, indicating a leaner carcass with less intramuscular fat. Additionally, the loin muscle area tended to be greater in castrated animals, suggesting a potential increase in muscle mass. These findings align with expectations, as castration reduces testosterone levels, which can promote fat deposition and limit muscle growth. While dressing percentages (slaughter weight as a proportion of live weight) were similar between rams and wethers, the meat quality analysis revealed a notable difference. Castrated sheep produced meat with increased instrumental tenderness. This suggests that castration might contribute to a more desirable eating experience for consumers who prefer softer, more palatable meat. This finding aligns with established knowledge, as castration reduces testosterone levels, leading to a decrease in muscle breakdown and potentially an increase in protein synthesis.

### 3.2. Genetic Background of Meat Production and Quality in Sheep

The analysis of QTL data identified several genomic regions linked to economically important meat production traits. For instance, a QTL on chromosome 20 was consistently associated with an increased marbling score, a desirable carcass quality attribute. Finally, transcriptome analysis revealed differential expression patterns of genes involved in muscle growth and development pathways in sheep with superior muscle characteristics. Genes associated with myogenesis, protein synthesis, and metabolic regulation were upregulated in these animals, suggesting their potential role in driving muscle growth and development.

Furthermore, we found different compilations of the top QTLs identified in sheep, specifically associated with meat production and quality traits, as cataloged in the Animal QTL Database updated to the years 2022–2023. QTLs represent genomic regions associated with the variation of quantitative traits. In this context, the focus is on traits relevant to meat production and quality, including but not limited to traits such as carcass weight, loin muscle depth, intramuscular fat content, meat color, growth in the Sheep Genome, fatness, pH, anatomy, and tenderness. 

#### Top Sheep QTLs Associated with Meat Production and Quality in the 2022–2023 Database

The variations in QTL numbers across different traits underscore the complexity of genetic regulation in sheep and provide valuable information for breeding programs aimed at improving meat production and quality in sheep populations [17]. The results from Table 3 displaying the top sheep QTLs associated with meat production and quality shed light on the intricate genetic underpinnings that influence various traits in sheep. The high numbers of QTLs associated with body dimensions such as tail fat deposition, backfat thickness, and heart girth emphasize the genetic complexity underlying these physical characteristics in sheep. This indicates that selective breeding strategies targeting these specific QTLs could potentially lead to improvements in meat production and quality [17]. Moreover, the significant number of QTLs associated with traits related to weight, production, and quality further underscores the genetic control over important aspects of sheep farming. Traits like weaning weight, birth weight, and various body weight measurements at different stages of growth highlight the genetic variability in growth patterns and production efficiency among sheep populations. Understanding and utilizing these QTLs in breeding programs could facilitate the development of sheep breeds with improved growth rates and higher-quality meat [38]. In the present investigation, the numbers of QTLs for many species compared to sheep based on the Animal QTL Database updated to 2022–2023 with a spotlight on meat quality are shown in Table 3, Figure 2, and Appendix A. Also, the curated data for different methods/marker types in sheep are summarized in Appendix A.

QTLs/associations for various meat carcass and quality traits in sheep are shown in Appendix A, and include meat sensory characteristics (Appendix A), meat carcass traits/anatomy (Appendix A), chemistry/pH (Appendix A), fatness (Appendix A), fat acid content (Appendix A), meat color (Appendix A), meat texture (Appendix A), and growth and carcass characteristics (Appendix A). 

Furthermore, the data highlight the importance of genetic factors in influencing traits related to sheep growth, fatness, production, and meat quality. Traits like body weight, average daily gain, and weaning weight had high numbers of associated QTLs, reflecting the genetic complexity underlying these characteristics [58]. Additionally, QTLs associated with meat and carcass traits were also prominent, emphasizing the genetic control over important meat quality attributes in sheep. 

### 3.3. Functional Enrichment Analysis

To decipher the main biological mechanisms or pathways affected by the 500 DEGs, which include 221 upregulated and 279 downregulated genes in fat-tailed and thin-tailed sheep breeds, a functional enrichment analysis was conducted. None of the upregulated genes, indicating increased expression in fat-tailed breeds, reached significance at a False Discovery Rate (FDR) threshold of <0.05 (Figure 3).

The investigated enriched terms among the DEGs were those associated with muscle growth and development processes (Appendix A) and meat quality (Appendix A), and the most enriched terms among the DEGs were associated with fat metabolism (Figure 3), encompassing both GO terms and KEGG pathways. This enrichment suggests a strong correlation between these genes and the process of fat-tail formation. Within these DEGs, a significant proportion were associated with terms related to fat metabolism, indicating their potential involvement in this biological pathway. Noteworthy genes encompassed within this category include *ELOVL6*, *PPP3CA*, *PDGES*, *TPI1*, *PLCB1*, *AMPD3*, *ND2*, *FASN*, *GADL1*, *COL6A3*, *PLCD1*, *DMT3A*, *SDS*, *LDHA*, *HK2*, *PGM2*, *GSTM3*, and *ACLY*. The majority of these genes either cleared at least 50% of the Jackknife tests or were located within the previously identified QTLs linked to sheep fatness. Certain genes within this subgroup warrant specific attention due to their prior associations with fat deposition. 

The formation of PPI networks through the STRING database corroborated that both up- and downregulated DEGs were integrated within functional interaction networks (Appendix A). It is a well-established fact that genes sharing a sub-network within a PPI network are likely to have similar functions and participate in similar biological processes. To investigate this further, a sub-network analysis was performed on the PPI network outcomes, revealing that two of the three identified sub-networks among the downregulated DEGs showed significant enrichment in various GO terms and KEGG pathways.

Hence, it is plausible to suggest that genes within the green sub-network may contribute to fat deposition in sheep tail breeds, given the notable enrichment of terms such as biosynthesis of unsaturated fatty acids (UFAs) and lipid biosynthetic process (LBP).

Yet, a collective of 30 biological processes and 6 KEGGs exhibited significant enrichment among the downregulated DEGs, indicating decreased expression in fat-tailed breeds, with an FDR < 0.05 (Figure 4).

Particularly noteworthy are the significant GO terms associated with fat metabolism, encompassing the “unsaturated fatty acid biosynthetic process” and the “fatty-acyl-CoA biosynthetic process” within the downregulated DEGs (Figure 3). Certain genes within this sub-network merit special attention due to their potential significance, such as *ACSF3*, *FASN*, *ACSL1*, *PPT-1*, and *HSD17B4*. These genes are situated within the regions of fatness-related QTLs or have successfully cleared at least 50% of the Jackknife tests (Appendix A). In a previous study, it was suggested that the downregulation of genes related to lipid metabolism in fat-tailed breeds could be associated with pathways extending beyond fat deposition, such as fat composition or fatty acid oxidation [12]. The literature indicates that the breed exerts a considerable influence on the fatty acid composition of tail fat.

In alignment with this, our current findings echo similar outcomes, as certain constituents of the green sub-network are engaged in pathways related to fat composition or fatty acid oxidation, including *CPTC1*, *ACFC3*, *ACSL1*, *ELOVL6*, and *SLC27A2*.

The *ELOVL6* gene, identified as a crucial regulator of fatty acid composition in mammals, assumes a central role in this metabolic pathway [59]. The initial step of de novo lipogenesis begins with the transformation of citrate into acetyl-CoA, facilitated by the enzyme ACLY. Acetyl-CoA, the subsequent molecule formed, plays a crucial role as a building block for various biochemical processes, including the synthesis of long-chain fatty acids, cholesterol, and histone acetylation [60]. Research has shown that adipocytes lacking ACLY tend to accumulate lipids in vivo and display discrepancies in both fatty acid levels and synthesis. In mammals, three ACSl1 isoforms are typically expressed, with their functions being evolutionarily conserved [61]. These genes play vital roles as transcriptional coregulators in the nucleus, enhancing the expression of genes involved in fatty acid oxidation [62]. 

Explorations into the functions of Fatty-Acid-Synthase (*FASN*), Acyl-CoA Synthetase Long-Chain-Family-Member 1 (*ACSL1*), and Palmitoyl-protein thioesterase-1 (*PPT-1*) genes in sheep fatty acid metabolism have provided significant insights. *FASN* is a fundamental enzyme in de novo fatty acid biosynthesis [40]. Studies in sheep have underscored its importance in governing fat deposition and composition, as it affects traits like marbling and adiposity [63]. *ACSL1* performs a crucial role in fatty acid metabolism through the catalysis of acyl-CoA production, a pivotal process in fatty acid activation for various metabolic pathways, such as oxidation and complex lipid synthesis [64]. Investigations in sheep have revealed correlations between *ACSL1* expression levels and rates of fatty acid oxidation, indicating its role in modulating energy metabolism and lipid utilization [65]. *PPT-1* is an enzyme involved in the breakdown of thioester linkages in palmitoylated proteins, potentially influencing various cellular activities, including lipid metabolism [65]. Although specific studies focusing on PPT1 in sheep fatty acid metabolism are limited, research in other species suggests its potential involvement in regulating lipid balance and cellular signaling pathways [66]. Conversely, specific pathways relevant to fat metabolism, such as extracellular matrix (ECM) receptor interaction, focal adhesion, and the PI3K-Akt signaling pathway were found to be significantly enriched within the pink sub-network, shedding light on the mechanisms underlying sheep fat-tail development. Noteworthy genes associated with these pathways including *COL1A1*, *SPP1*, *ITGA1*, *VEGFA*, *ITGA2*, *COL4A6*, *COL6A6*, *PDGFRA*, *SDC1*, and *COL6A3* were either situated within QTLs linked to fatness or successfully passed a minimum of 50% of the Jackknife tests (Appendix A), indicating their probable involvement in fat-tail morphogenesis. Focal adhesion, closely associated with the ECM, functions as a physical connection to the ECM. As a result, the differentially expressed genes (DEGs) linked to these two pathways exhibited notable similarities (Appendix A). 

The ECM is crucial for keeping tissues in balance, with a variety of important molecules found within it in adipose tissue. These include glycosaminoglycans, collagen, elastin, fibronectin, and laminin [67]. The interaction between the ECM and cell surface molecules, like integrins, plays a crucial role in regulating various cellular processes including growth, motility, and apoptosis [68]. Interestingly, the signaling cascade involved in ECM receptor interaction has been recognized as a significantly enriched pathway in DEGs between fat- and thin-tailed sheep breeds in various previous investigations, as well as in DEGs between lean and obese individuals [69]. Moreover, research has demonstrated that the communication between ECM components and transmembrane receptors of adipocytes is associated with site-specific adipogenesis in cattle [70].

Aligned with our discoveries, prior studies have intimated a potential correlation between the function of *ACSL1* and fatty acid oxidation specifically within the bovine hepatic system [12]. Also, the solute-carrier-family-27-member 2 (*SLC27A2*), functioning as a transmembrane transporter coenzyme, takes part in the beta-oxidation of long-chain fatty acids [71] and performs a crucial role in the degradation of fatty acids [72]. Noteworthy is the identification of *SLC27A2* as closely linked to the tail characteristics as observed in the investigation conducted that delved into transcriptome profiles associated with adipose accumulation in the tails of sheep [73]. Nevertheless, we argue that the animal science sphere has yet to comprehensively explore the involvement of the ECM receptor interaction pathway in fat metabolism and the development of fat tails in sheep. Fascinatingly, the ECM pathway was recognized as inhibited in the KEGG pathway examination as human mesenchymal stromal cells transformed into adipocytes [74]. The upregulation of genes in this pathway, such as collagen subunits IV, V, and VI (*COL1A1*, *COL6A3*, *COL4A6*, and *COL6A6*), multiple integrins (*ITGA-1* and -*2*), and proteoglycans such as Syndecan-1 and -4 (*SDC-1* and *-4*), aligns with our research findings.

The weakened activity of these genes is tied to the reconstruction of the cytoskeleton as adipocytes are being produced. Notably, the breakdown of collagen type IV has been associated with the differentiation process towards adipogenesis. 

A comprehensive analysis focusing on the function of *COL6A3* in the context of obesity and diabetes in the human population revealed an increase in *COL6A3* expression following weight reduction, exhibiting a negative relationship with obesity, which is consistent with our current research findings [75]. Therefore, it is reasonable to infer a potential correlation between these genes and the development of fat tails. A total of 100 annotated QTLs linked to fatness were extracted from the Sheep Animal QTLdb database [40]. 

A total of 113 genes, comprising 50 genes with increased expression and 63 genes with decreased expression (Figure 4 and Figure 5), were significantly (*p*-value = 0.018) located within the genomic coordinates of 16 QTLs (Figure 4). These QTLs were spread across chromosomes 1, 6, 8, 10, 11, 12, 14, 16, 18, 19, and 23, associated with various traits including internal fat content, carcass fat weight, carcass fat percentage, muscle depth at the third lumbar, dressing percentage, subcutaneous fat area, subcutaneous fat thickness, carcass length, and lean meat yield percentage. Among the 113 genes, 35 upregulated and 55 downregulated genes (DEGs) were consistently identified in at least 50% of the Jackknife test iterations (≥80%). Subsequently, an analysis was conducted to assess the concentration of DEGs within the QTLs, conducted separately for the DEGs from each study. Notably, among the six studies examined, only the initial study’s DEGs exhibited significant enrichment within the QTLs.

### 3.4. PPI Network Analysis

To establish a PPI network of DEGs comprising 500 genes, with 221 showing upregulation and 279 exhibiting downregulation in fat- versus thin-tailed sheep breeds, we employed the STRING database. Initially, ENSEMBL accession numbers were assigned to the DEGs, resulting in the annotation of 361 gene names (135 upregulated and 226 downregulated). Consequently, a total of 125 genes and 218 genes were successfully identified and utilized to create the PPI network for the up- and down regulated DEGs, respectively.

The final PPI network of upregulated DEGs consisted of 72 nodes interconnected by 98 edges, signifying known or predicted interactions (PPI enrichment *p*-value = 0.005). Moreover, the gene network interaction among the downregulated DEGs consisted of 157 nodes linked by 337 edges, with a significant PPI enrichment *p*-value of <1.0 × 10^−16^ (Figure 5).

### 3.5. PPI Network of Top 100 Genes Associated with Growth Muscle 

The resultant PPI networks underwent module analysis using a k-means clustering approach, revealing three significant sub-networks within each network. Specifically, in the upregulated DEGs, three sub-networks were identified: the orange sub-network comprised 22 nodes and 20 edges (*p*-value = 6.51 × 10^−9^), the yellow sub-network contained 24 nodes and 55 edges (*p*-value = 5.49 × 10^−11^), and the light-blue sub-network consisted of 20 nodes and 16 edges (*p*-value = 2.89 × 10^−6^) (Figure 4). The functional enrichment assessment of the clusters revealed six notable biological processes in the orange sub-network, encompassing terms such as gene expression, RNA metabolic process, and translation. Conversely, in the downregulated DEGs, the green sub-network comprised 34 nodes and 41 edges (*p*-value < 1.0 × 10^−16^), the pink sub-network contained 64 nodes and 90 edges (*p*-value < 1.0 × 10^−16^), and the purple sub-network consisted of 51 nodes and 105 edges (*p*-value < 1.0 × 10^−16^) (Figure 6). Significantly, genes in the green sub-network showed notable enrichment in the LBP biological process (FDR < 0.04) and the biosynthesis of UFAs KEGG pathway (FDR < 0.01). Additionally, the pink sub-network highlighted 17 biological processes (such as ECM and anatomical structure morphogenesis) and 11 KEGG pathways (including ECM receptor interaction, PI3K-Akt signaling pathway, and focal adhesion). Conversely, no significant terms were identified within the purple sub-network.

## 4. Conclusions

The meta-analysis conducted integrated multiple datasets to explore the complex relationship among castration, QTLs, and DEGs in sheep muscle development. The study revealed the significant impact of castration on various aspects of sheep physiology, particularly on muscle development and related characteristics. By confirming the influence of castration on gene expression associated with fat metabolism and muscle growth, the importance of considering castration status in sheep muscle research was emphasized. Insight into QTLs linked to muscle growth was also provided, identifying genetic foundations and potential targets for enhancing meat quality through breeding programs. Functional interactions among DEGs were verified using PPI network analysis, highlighting associations with key pathways and molecular mechanisms related to fat metabolism and muscle development. Overall, this meta-analysis offers a comprehensive understanding of the factors influencing muscle growth in sheep, emphasizing the significance of genetic and molecular mechanisms in optimizing meat production and quality in sheep populations.

## Figures and Tables

**Figure 1 animals-14-01679-f001:**
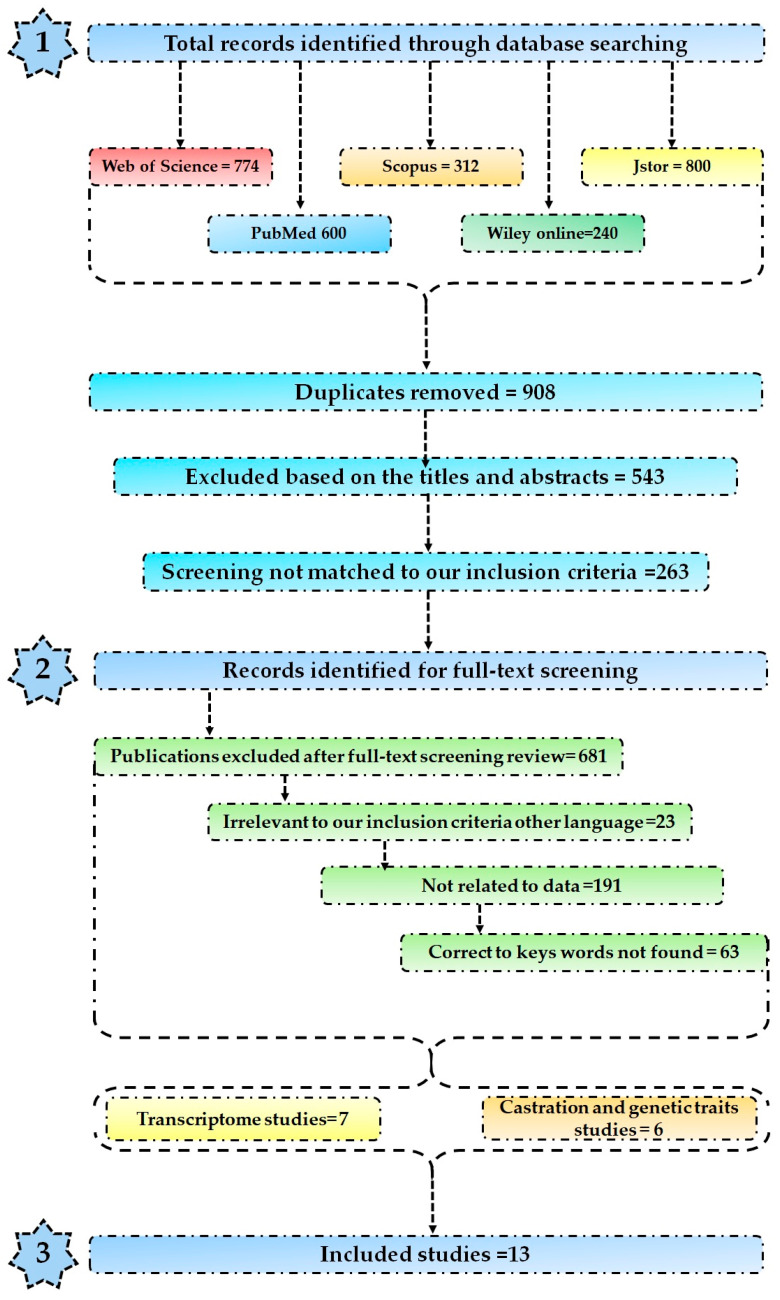
Flowchart of the different databases used for data collection.

**Figure 2 animals-14-01679-f002:**
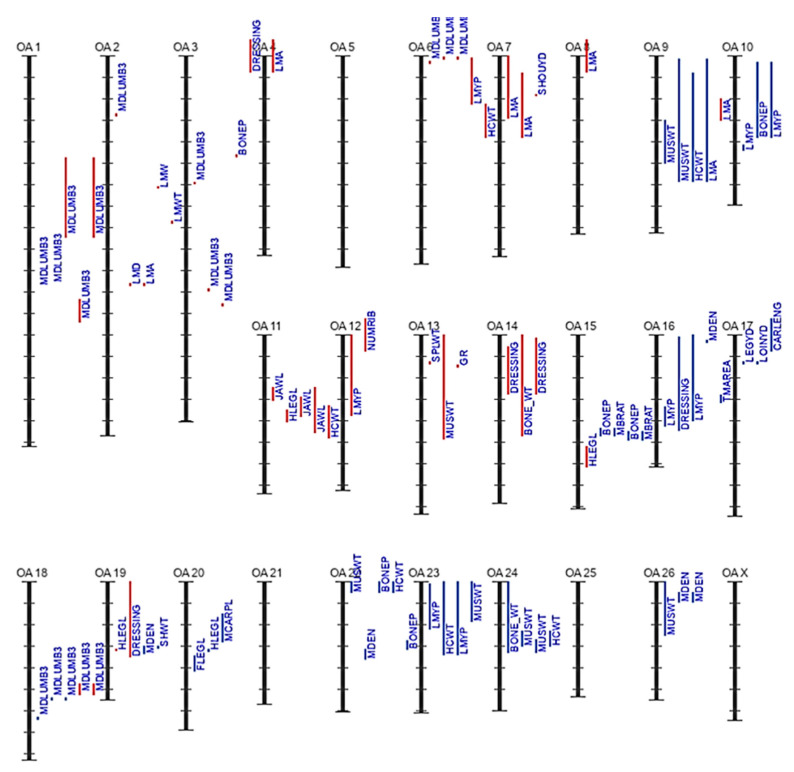
QTLs/associations for meat carcass traits (anatomy) in the Sheep Genome (it shows genome locations where anatomy traits are mapped by QTLs or SNP associations).

**Figure 3 animals-14-01679-f003:**
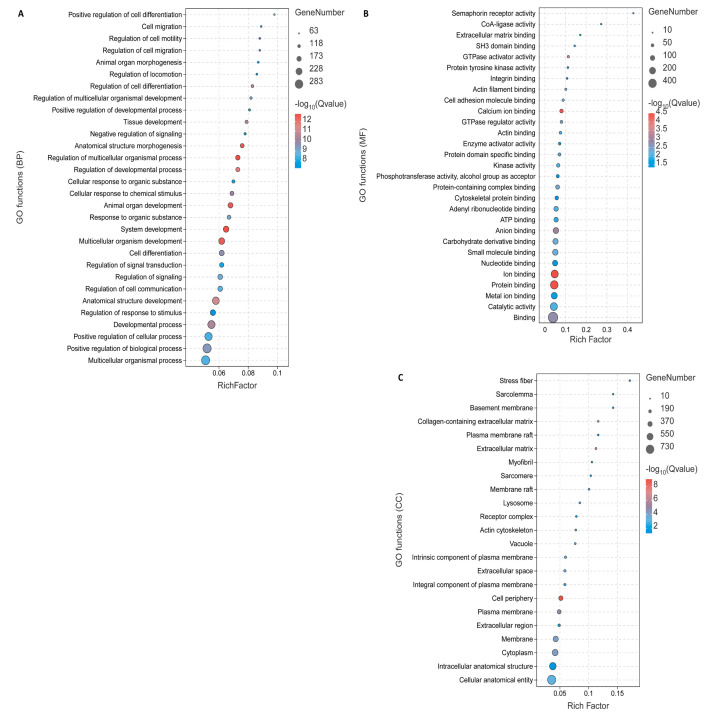
The enriched Gene Ontology (GO) terms associated with the functions of DEGs were identified. (**A**) A bar chart representing the top 30 biological process (BP), (**B**) cellular component (CC), and (**C**) molecular function (MF) terms from the enrichment analysis of DEGs in the growth muscles.

**Figure 4 animals-14-01679-f004:**
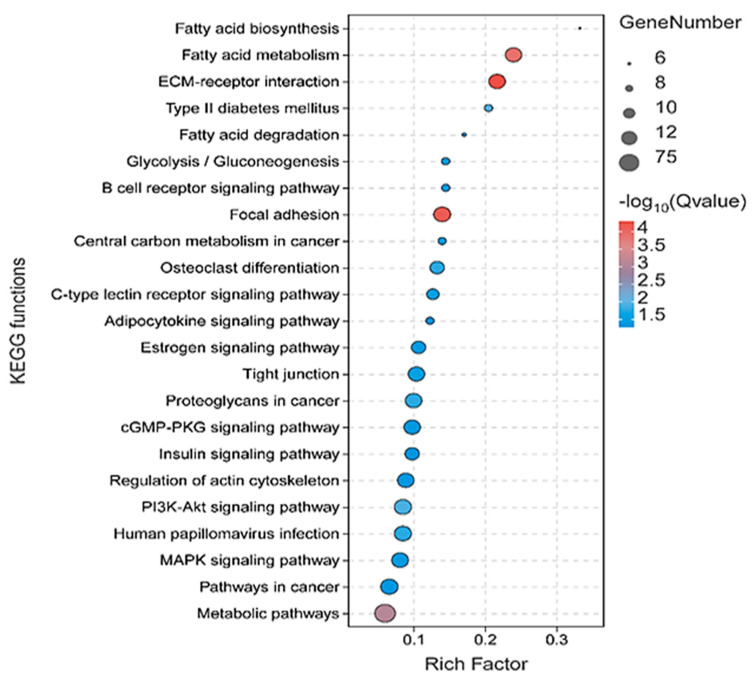
Bubble plots illustrating the functional pathways enriched in the KEGG analysis. The top 23 pathways are presented, showcasing both the upregulated and downregulated genes associated with muscle growth pathways.

**Figure 5 animals-14-01679-f005:**
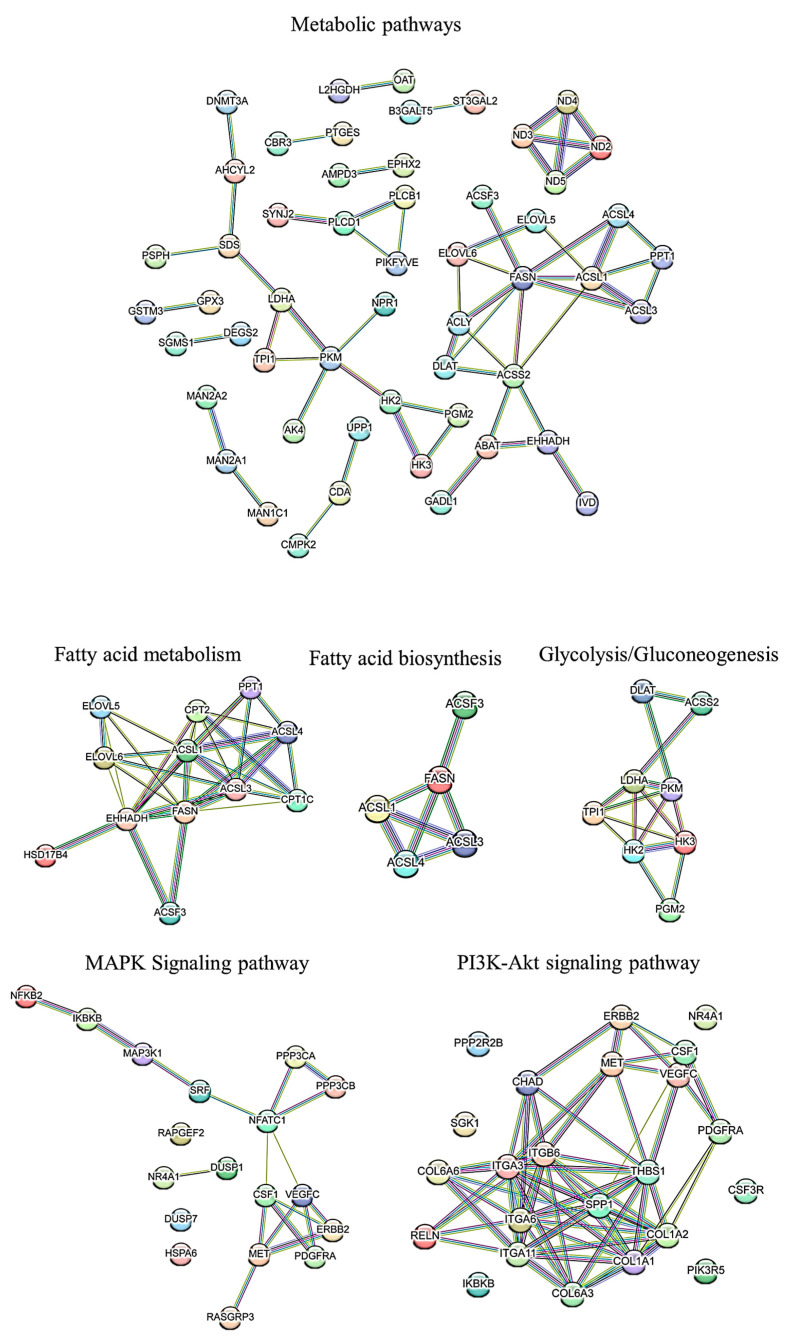
The PPI network and functional cluster of DEGs are associated with metabolic pathways and fatty acid pathways. The circles in the figure represent DEGs found in approximately half of the Jackknife test results, with orange, yellow, and light red circles indicating differentially expressed genes related to these pathways.

**Figure 6 animals-14-01679-f006:**
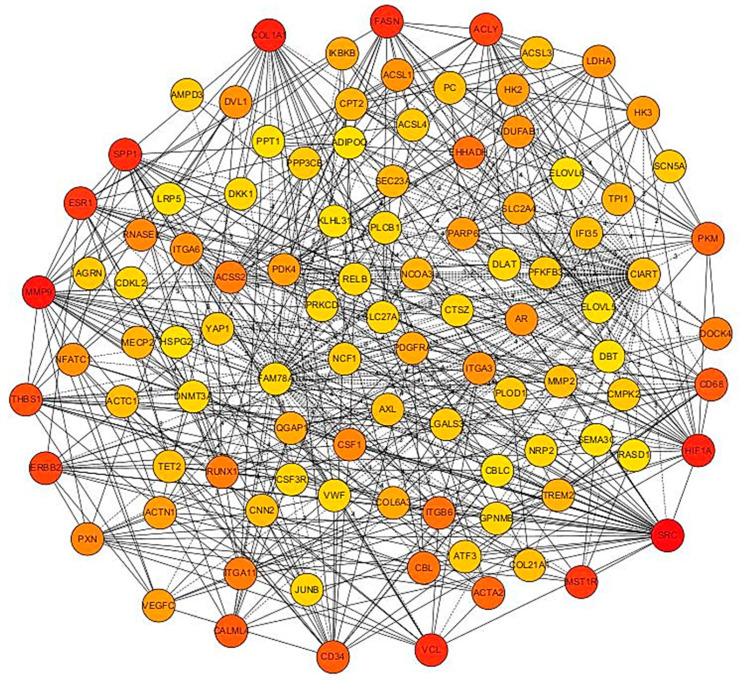
The PPI network and functional clusters of the top 100 genes associated with growth muscle. The PPI networks underwent module analysis using k-means clustering, revealing significant sub-networks within each network. In the upregulated DEGs, three sub-networks were identified: orange (22 nodes, 20 edges), yellow (24 nodes, 55 edges), and red (20 nodes, 16 edges).

**Table 1 animals-14-01679-t001:** List of the comparative transcriptome studies on the fat- and thin-tailed sheep breeds.

Fat (Breeds) ^1^	Thin (Breeds) ^2^	Number of Replicates	Upregulated Genes ^3^	Downregulated Genes ^4^	Type of Clean Reads ^5^	Gender ^6^	References
STH ^7^	PD ^8^ *×* STH	100	457	417	10,278,304,671	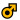	[21]
QHMM ^9^	Merino *×* STH	60	405	555	47,441,397	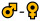	[22]
Lori-Bakhtiari	Zel × Lori-Bakhtiari	6	80	184	150 bp	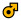	[12]
Wagyu	Hostein × Wagyu	98	321	341	129,097,834	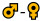	[23]
PD versus	STH× PD versus	21	33	32	120 bp	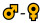	[24]
India Bandur	Bandur	20	98	43	93,434,064	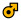	[27]
Gayal	Mithun/Gayal	4	173	124	241,738,022	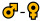	[25]

A catalog of sheep breeds exhibiting the following: ^1^ Fat-tailed sheep; ^2^ Thin-tailed sheep, accompanied by the number of biological replications; ^3^ The identification of genes that are upregulated and ^4^ downregulated in sheep breeds with fat tails; ^5^ The clean read data of transcript; ^6^ Gender, 
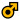
: Male and 
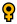
: Female; *^7^* STH, Small Tail Han; ^8^ PD: Polled Dorset; ^9^ QHMM, Qianhua Mutton Merino sheep.

**Table 2 animals-14-01679-t002:** The essential characteristics of the studies incorporated in the present meta-analysis investigating the effects of castration on diverse aspects of sheep production. The table provides information on breed, method of castration, age of castration, slaughter age/weight, and response variables.

Breed	Method of Castration	Age of Castration	Slaughter Age/Weight	Rams	Wethers	Response Variables	Country	Reference
MSB ^1^ × BB ^2^	RRi ^9^		40 kg	18	18	IMF	Mexico	[41]
Mongolian sheep		3 m ^10^	3 m/40 kg	3	4	CW	China	[42]
STS ^3^ × DP ^4^	RRi	11 m	11 m/39 kg, 64 kg	2	2		China	[43]
QHMM ^5^ × STH ^6^		1 y ^11^	1 y/55 kg, 45 kg	30	30	CW	China	[44]
Bandur × local sheep	-	2 m	6 m/38 kg	16	-	CW	India	[45]
		40 h ^12^	8 m/49.86, 40.64 kg	3	-		China	[46]
Native sheep	surgical	5 d ^13^	45 kg	47.6	12	DM, CP	Bangladesh	[47]
SB ^7^ × TXSB ^8^	RRi	48 h	64 kg, 66 kg	16.50	17.25	DM	Ireland	
SB × TXSB	RRi	50 h	20.8 kg, 23.3 kg	50	50	IMF	Ireland	[48]
Merino	RRi	11 weeks	20.7	30	-	CW	Australia	[47]
Washera sheep	RRi	1 m 15 d	1 y/53.3, 53.9 kg	12	12	CW, DP	Ethiopia	[49]
Fabrianese	-	5 d	23, 37.4, 41	14	14	DM	Italy	[50]
White Dorper	-	24 h	34.4	6	6	DM	Australia	[51]
Dhamari sheep	-	14 d	23.7	3	3		Yemen	[52]
Boer goat	-	-	30.4	32	-	DM	Brazil	[53]
Doper lambs		-	40.17	45	-	DM	Mexico	[54]
Santa Ines × Dorper	-	-	28 d/31 kg	72	36	DM, CP, NDF	Brazil	[55]
Awassi sheep	-	-	4 m/43.5 kg	12	-	CW	Qatar	[56]
Black goat	RRi	24 h	20 d/25 kg	-	24	DG, REA	Iraq	[57]

^1^ MSB, Mexican Sheep breed; ^2^ BB, Black Belly; ^3^ STS, Small Tail Sheep; ^4^ DP, Dorper; ^5^ QHMM, Qianhua Mutton Merino sheep; ^6^ STH: Small Tail Han; ^7^ SB, Scottish Blackface; ^8^ TXSB, Texel × Scottish Blackface; ^9^ RRi, rubber ring; ^10^ m, month; ^11^ y, year; ^12^ h, hours; ^13^ d, days.

**Table 3 animals-14-01679-t003:** This table provides insights into the key sheep quantitative trait loci (QTLs) associated with meat production and quality in the database covering the years 2022 to 2023.

No.	Traits		Number of QTLs
3	Body dimensions	Tail fat deposition	64
4		Body length	8
5		Chest width	7
6		Longissimus muscle area	7
7		Shin circumference	7
8		Backfat thickness	18
9		Heart girth	18
10		Body weight at hatch	17
11		Withers height	17
12		Chest depth	16
13	Weight, Production and Quality	Body weight	279
14		Average daily gain	95
15		Weaning weight	4464
16		Birth weight	4185
17		Body weight at 6 months	2232
18		Yearling weight	1674
19		Body weight at 8 weeks	1116
20		Body weight at 20 weeks	1116
21		Body weight at 180 days	837
22		Body weight at 9 months	837
23		Litter size (first parity)	708
24		Slaughter weight	558
25		Adult weight	558
26		Growth	589
27		Fatness	174
28		Production	610
29		Meat and carcass	563

## Data Availability

All data generated or analyzed during this study are included in the article and its main and Appendix A.

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
