# Peer review of "Integrative Meta-Analysis: Unveiling Genetic Factors in Meat Sheep Growth and Muscular Development through QTL and Transcriptome Studies"

_animals, 2024, doi:10.3390/ani14111679_

Round 1

Reviewer 1 Report

Comments and Suggestions for Authors

Question 1: Please rewrite the Abstract section in the order of "Objective, Methods, Results, and Conclusions.

Question 2: line 82,299,416, and 495 you should unified references in your manuscript, please check it carefully.

Question 3: In Tables 1 and 2, the reference section must be revised, and make sure about table 3 fond size is not cleared.

Question 4: In the figures 5 and 6 legend, the descriptions of “1,2” and “3” are separated by colours.  Please check it.

Question 5: In the Results and Discussion section, some explanations needed scientific supports.

Question 6: the logic in these lines 611 – 615 should be revised.

Question 7: the repetition of this sentence “presented in at least 50% of the Jackknife test” seem not good. Please revised it.

Question 8: I found some of references are not in the same format please be revised carefully.

Question 9: In the Results and Discussion section, some explanations needed scientific supports

Question 10: The conclusion section is too long. Please summarized it.

Author Response

Please see the attachment/Kindly see the attached PDF file, it contains ''Point-by-Point Response to Reviewers and Editors''. Thanks for your helpful comments.

Reviewer 2 Report

Comments and Suggestions for Authors

Review on the manuscript titled “Integrative Meta-Analysis: Unveiling Genetic Factors in Meat Sheep Growth and Muscular Development through QTL and Transcriptome Studies” by Rekhman et al., 2024.

                The ultimate goal of the manuscript is to elaborate on the castration impact on the metabolic networks of the sheep “

                First, the authors compiled the published studies on the subject “examining sheep carcasses raised under various production systems and across diverse geographical region(s)”. After processing the big volume of data on their specific query (Fig.1), they were left with 6 studies. RNA-seq data were used for further analysis.

They present basic RNA-seq stats in Table 1, but there are no # replicates, and the abbreviations in the table column titles are quite incomprehensive, along with stars per each column titles: not all of stars sets are explained in references below the table. Also, looks like the table should be placed lower in the text.

                Chapter 2.3. presents RNA-Seq assembly routine. Next (2.4) follows rather chaotic description of what’s apparently is a batch effect fixing. It is presented in non-canonical form regarding to formulas presentation. Also, one could wonder why not using a standard routine in R package for this purpose.

Chapter 2.4. is not immediately comprehensive due to abundance of non-disclosed abbreviations.

There is no 2.5. Chapter. Chapters 2.6.-2.9 present RRAs/VCR methods as a filtering routine.

Chapters 2.10-2.11 devoted to annotation methods and 2.12. describes QTL routine.

The authors describe consequent QTL methods applications with certain inferences. The results are provided in Table 3, underlining the enrichment of QTLs to ‘tail fat deposition’ trait (64).

Chapter 3.2.1. devoted to the castrated samples of data which was also employed into target traits related QTL mapping analysis. They proved the QTL genomic map  and report association QTLs to the genes related for target traits.

Chapter 3.3. is devoted to functional enrichment analysis. They provide Figure 3, 4. Bubble plots for DEGs underlining essentially enriched KEGG categories. As they observed high dependence of target traits with fat/thin tails, they compared the corresponding DEGs groups in chapter 3.4 for the enrichment employing sring-db.org (Fig. 5).

                Lastly, the present PPI network for the top 100 DEGs associated with Muscle growth rate (Fig. 6).

                Overall, the authors performed a profound work, and discussed several groups of genes on their possible implications in the target traits. Still, I find manuscript not mature enough due to multiple inconsistencies rendering a thoroughly revising session.

I found no Supplementary materials available.

Some notes are presented below.        

1)      The authors claim that they target the effects of castration in their study, but major trait considered is a tail’s fat state. It’s not immediately clear if tail state is related to castration.

2)      Table 1 is profoundly incomprehensive. While one can see 6 studies, there are no replicates number per study. The superscript below the table claims: “Table 1:A catalog of sheep breeds exhibiting fat and thin tails, accompanied by the quantity of biological replications (???) *** the identification of genes that are up-regulated and down-regulated in sheep breeds with fat tails.**** the clean read data of transcript.” Please clarify, and leave only star marked endnotes under the table, all other information merge to the table title.

3)      Fig. 1. WilEy online, Also, font the size/quality is inappropriate.

4)      “The acritical focused on studies examining sheep carcasses raised under various production systems and across diverse geographical regions” I’m not sure ’acritical’ is relevant to this sentence.

5)      P.4 s181“As a result, the final dataset comprised 16 studies extracted from these seven research papers. “ … P.5 s192: “Therefore, only six datasets were suitable for our analysis.”

6)      Fig 5 subscript: “Oval circles indicate the DEGs that were presented in at least 50% of the Jackknife test results” – no ovals therein.

7)      Figure 6: “from sp(T)ring database”; “Orange, yellow and light red nodes represent orange, yellow and light blue sub-networks, respectively.” – what does it mean? There is no light-blue shadings in the picture?

8)      More on Fig 6 subscript: “Oval circle Oval circles indicate the DEGs that were presented in at least 50% of the Jackknife test results. The DEGs that were located in QTL regions related to fatness are represented as bold texts.” – There are no ovals in Fig. 6. Probably the figure is missing some parts, or it’s related to another one (Fig.4)?

Comments on the Quality of English Language

English should be addressed in multiple instances.

Author Response

(The authors gave the same response as above.)

Round 2

Reviewer 2 Report

Comments and Suggestions for Authors

The authors did a massive correction addressing my comments.

Some notes are given below.

1)      Fig. 1: ‘WilEy’ should be replaced with Wiley, since I capitalized the ‘e’ letter for identification purposes only.

2)      Figure 5. Gene names should be shifted a bit aside from the nodes circles since they are unreadable this way (optional).

3)      Table 6: It would be rational to order genes by log2fold from max to min.

4)      Table. S7: a) there are 9 NTU gene abbreviations with different log2 scores. Please address its meaning in the Table title legend. Also, please report what does ‘---‘ stand for? If it corresponds to ratio containing null in upper/lower part, it still can be assessed as –inf, inf (at least a sign). If that’s something else , then it should be explained in the table title B) It’d be good to range the genes according to log2fold score max->min.

5)      Table S8 please convert the digital format to floating point one, with at most 3-4 mantissa digits for transparency.

Author Response

Please see the attachment. Kindly See ''The Point-by-Point Response file''
